# Metabolic and Oxidative Stress Management Heterogeneity in a Panel of Breast Cancer Cell Lines

**DOI:** 10.3390/metabo14080435

**Published:** 2024-08-06

**Authors:** Paola Maycotte, Fabiola Lilí Sarmiento-Salinas, Alin García-Miranda, Cesar Ivan Ovando-Ovando, Diana Xochiquetzal Robledo-Cadena, Luz Hernández-Esquivel, Ricardo Jasso-Chávez, Alvaro Marín-Hernández

**Affiliations:** 1Centro de Investigación Biomédica de Oriente, Instituto Mexicano del Seguro Social, Km 4.5 Carretera Atlixco-Metepec HGZ5, Puebla 74360, Mexico; lilith_spp@hotmail.com (F.L.S.-S.); biol.aligami@gmail.com (A.G.-M.); 2Consejo Nacional de Humanidades, Ciencia y Tecnología (CONAHCYT), Mexico City 03940, Mexico; 3Laboratorio de Biología Celular del Cáncer, Universidad Autónoma de Guerrero, Chilpancingo de los Bravo 39090, Mexico; 4Departamento de Bioquímica, Instituto Nacional de Cardiología Ignacio Chávez, Mexico City 14080, Mexico; cesar.ovando.ovando@gmail.com (C.I.O.-O.); xochiquetzal@ciencias.unam.mx (D.X.R.-C.); maria.esquivel@cardiologia.org.mx (L.H.-E.); rjassoch@gmail.com (R.J.-C.)

**Keywords:** breast cancer, energy metabolism, reactive oxygen species, antioxidant system

## Abstract

Metabolic alterations are recognized as one of the hallmarks of cancer. Among these, alterations in mitochondrial function have been associated with an enhanced production of Reactive Oxygen Species (ROS), which activate ROS-regulated cancer cell signaling pathways. Breast cancer is the main cancer-related cause of death for women globally. It is a heterogeneous disease with subtypes characterized by specific molecular features and patient outcomes. With the purpose of identifying differences in energy metabolism and the oxidative stress management system in non-tumorigenic, estrogen receptor positive (ER+) and triple negative (TN) breast cancer cells, we evaluated ROS production, protein enzyme levels and activities and profiled energy metabolism. We found differences in energetic metabolism and ROS management systems between non-tumorigenic and cancer cells and between ER+ and TN breast cancer cells. Our results indicate a dependence on glycolysis despite different glycolytic ATP levels in all cancer cell lines tested. In addition, our data show that high levels of ROS in TN cells are a result of limited antioxidant capacity in the NADPH producing and GSH systems, mitochondrial dysfunction and non-mitochondrial ROS production, making them more sensitive to GSH synthesis inhibitors. Our data suggest that metabolic and antioxidant profiling of breast cancer will provide important targets for metabolic inhibitors or antioxidant treatments for breast cancer therapy.

## 1. Introduction

Metabolic reprogramming is acknowledged as an emerging hallmark of cancer [1,2] since adjustments in energy metabolism are often necessary to maintain the sustained proliferation of cancer cells. Cancer cells often increase their metabolism by enhancing transporter and enzyme levels, mediated by oncogene activation and divert metabolites towards biosynthetic pathways for nucleoside and amino acid production to sustain cellular proliferation [1]. Thus, cancer cells have been proposed to maintain a permanent anabolic state supported by the constituent activation of pro-survival and proliferation signaling pathways [3]. The initial observation by Otto Warburg that tumor cells use glycolysis to generate ATP even in the presence of oxygen, led to the hypothesis of irreversible respiratory damage in cancer cells [4]. However, much evidence indicates that tumor cells possess functional mitochondria capable of performing oxidative phosphorylation (OXPHOS), with important contributions to the cellular ATP supply, contributing to tumor cell maintenance and supporting the invasive capacity of cancer cells [3,4,5]. Indeed, activated oncogenes, like PI3K/AKT, HIF-1α, RAS or *c-myc*, as well as inactivated tumor suppressors including p53 and PTEN, are involved in cancer cell metabolic alterations even in the absence of extracellular signaling [6]. Therefore, many cancer cells and solid tumors exhibit enhanced glycolysis compared to normal cells [3,5], but with a limited contribution to the total cellular ATP production [5]. Glycolytic intermediates are diverted into the pentose phosphate pathway (PPP) to support ribose or NADPH production; and in general, to support biosynthesis [4]. Lactate, the product of anaerobic glycolysis, has been shown to acidify and favor a tumorigenic microenvironment, and extratumoral lactate has also been proposed to be used as a source of oxidizable carbon for cancer cells [4].

In addition to glucose metabolism, mitochondria are important ATP sources in cancer, and contribute to biosynthetic pathways. Increased glutamine uptake has also been described, mediated mainly by *c-myc* transformation or epidermal growth factor receptor (EGFR) signaling, and used for nucleotide, hexosamine or asparagine biosynthesis. Glutamine also fuels the tricarboxylic acid (TCA) cycle and modulates oxidative stress by providing glutamate for glutathione (GSH) biosynthesis [3]. In addition, TCA cycle intermediates can also be used for biosynthesis; and the SREBP1 transcription factor, an activator of lipid biosynthesis, or FASN fatty acid synthase, have been found to be upregulated in cancer cells with essential activities in tumorigenesis [3]. Importantly, OXPHOS plays a role as the main ATP supply in many cancer cells; NADH from the TCA cycle is a substrate for the electron transport chain (ETC), which generates the H^+^ electrochemical gradient necessary for ATP synthesis by ATP synthase. Also, ETC is necessary for the regeneration of oxidized NAD+ and FAD, for pyrimidine biosynthesis, for glutamine metabolism [3,5], for the production of reactive oxygen species (ROS), redox molecules or metabolites and the regulation of cellular signaling or cell death [7]. Thus, cancer energetic metabolism has been shown to directly contribute to malignancy traits and to promote cancer cell proliferation and invasion.

In order to cope with metabolic adaptation, cancer cells exploit diverse cellular mechanisms, including ROS regulating systems, autophagy or an increased nitrogen demand to comply with biosynthesis [3]. Increased ROS levels are a common feature in cancer and have been related to regulating cancer-promoting signaling pathways [6,8]. Thus, cancer cells need to increase their antioxidant protein and pathway levels to cope with high ROS. By using glutamine as a GSH precursor, glutathione peroxidase (GPx) oxidizes reduced GSH to oxidized glutathione (GSSG) and simultaneously reduces peroxides to water. The glutathione reductase (GR) then reduces GSSG to GSH oxidizing NADPH. Therefore, the generation of NADPH by the malic enzyme, NADP-dependent isocitrate dehydrogenase (IDH) or the PPP pathway (glucose 6-phosphate dehydrogenase, G6PDH; 6-phosphogluconate dehydrogenase, PGDH) is necessary for ROS detoxification by the GSH system (GPx/GR). Alternatively, cancer cells also cope with ROS by increasing NRF2 signaling and antioxidant enzyme expression (enzymes involved in GSH synthesis, thioredoxin, thioredoxin reductase, sulfiredoxin, peroxiredoxin, GPx, glutathione S-transferases, catalase (CAT) and superoxide dismutase (SOD)) [6,9,10].

Breast cancer is the main cancer-related cause of death for women globally. The purpose of this study is to identify differences in energy metabolism and the oxidative stress management system in non-tumorigenic and breast cancer cells. Our study evaluated a panel of breast cancer cell lines belonging to the estrogen receptor positive (ER+) or triple negative (TN) subtypes, with a particular focus on mitochondrial metabolism and ROS management systems to suggest specific metabolic vulnerabilities that could define them or be used as targets for therapy.

## 2. Materials and Methods

### 2.1. Cell Lines and Cell Culture

Non-cancer breast MCF10A cells were cultured in DMEM/F12 (Caisson Labs, Smithfield, UT, USA) supplemented with 5% horse serum (Biowest, Nuaillé, France), 0.5 μg/mL hydrocortisone (Sigma, St Louis, MO, USA), 20 ng/mL EGF (Preprotech, ThermoFisher Scientific, Waltham, MA, USA), 100 ng/mL cholera toxin (Sigma, St Louis, MO, USA), 1% penicillin/streptomycin (Sigma) and 10 μg/mL insulin (Sigma, St Louis, MO, USA). Breast cancer cell lines: MCF7 (ER+/luminal) was grown in Eagle’s MEM (Caisson Labs, Smithfield, UT, USA) with 10 μg/mL insulin, 1% penicillin/streptomycin and 10% fetal bovine serum (FBS) (Biowest, Nuaillé, France). T47D (ER+/luminal) was cultured in RPMI-1640 (Caisson Labs) with 7.5 μg/mL insulin, 1% penicillin/streptomycin and 10% FBS. MDAMB231 (TN) was grown in DMEM/F12 with 1% penicillin/streptomycin and 10% FBS. BT549 (TN) was cultured in RPMI-1640 (Caisson Labs) with 7.5 μg/mL insulin, 1% penicillin/streptomycin and 10% FBS. Cells were initially acquired from the University of Colorado Cancer Center Tissue Culture Core and were a kind gift from Dr. Andrew Thorburn. Genotyping (INMEGEN, Tlalpan, Mexico City, Mexico) of T47D and BT-549 used in the present study revealed that they are subclones of the original cancer cell lines, while MCF10A, MCF7 and MDAMB23 cell lines showed >90% of the canonic allelic markers displayed in the ATCC original clones.

### 2.2. ROS Measurement

ROS were evaluated by flow cytometry. Briefly, 100,000 cells were plated on 6 well plates and after 24 h stained with 10 μM dihydroethidium (DHE, Sigma Aldrich, D7008-10MG) in culture medium for 30 min at room temperature. Cytosolic DHE exhibits blue fluorescence; however, once this probe is oxidized to ethidium, its fluorescence shifts to bright red. This dye has been shown to be oxidized by superoxide anion but also by hydrogen peroxide [11], thus we used it to measure total ROS production. After incubation, cells were washed three times, first with pre-warmed complete medium and then with PBS, trypsinized, collected, and centrifuged at 2500 rpm. The pellet was resuspended in PBS with 3% FBS and filtered for immediate analysis in a BD FACS Canto II flow cytometer with an excitation wavelength of 488 nm and 670/LP, 655/LP emission filters [12].

### 2.3. Western Blot Assays

For total protein cell extraction, cells were mixed with lysis buffer RIPA (PBS 1X, pH 7.2, 1% IGEPAL, 0.1% SDS and 0.05% sodium deoxycholate) plus 1 mM PMSF (phenyl methane sulfonyl fluoride) and 1 tablet of complete protease inhibitors cocktail (Roche, Mannheim, Germany), previously dissolved in 10 mL RIPA buffer. Once the protein concentration was determined by the Lowry method [13], the samples (50 µg) were re-suspended in a loading buffer containing 10% glycerol, 2% SDS and 0.5 M Tris-HCl, 0.002% bromophenol blue, pH 6.8 plus 5% β-mercaptoethanol and loaded onto 10% polyacrylamide gel. Electrophoretic transfer to PVDF membranes (BioRad; Hercules, CA, USA) was followed by overnight immunoblotting at 4 °C with 1:1000 dilution antibodies for β-actin (SC-376421), CAT (SC-365738), CS (citrate synthase, SC-390693), GR (SC-133245), HKII (hexokinase II, SC-374091), SOD 2 (SC-130345, Santa Cruz Biotechnology TX, USA), SOD 1 (MA1105, Thermo Fisher, Waltham MA, USA), IDH2 (abx128113, Abbexa Ltd., Cambridge, UK), G6PDH (ab87230), PGDH (ab129199, Abcam, Cambridge, UK) and 1:500 dilution antibodies for GPx1 (SC-22146, Santa Cruz Biotechnology, TX, USA) and GPx4 (PA5-19695, Thermo Fisher, Waltham MA, USA). The hybridization bands were revealed with the corresponding secondary antibodies conjugated with horseradish peroxidase (A16029 and 31320, Thermo Fisher, Waltham MA, USA; SC-2768, Santa Cruz Biotechnology, TX, USA). The signal was detected by chemiluminescence using the ECL-Plus detection system (Sigma-Aldrich MO, USA). Densitometry analysis was performed using the Scion Image Software Beta 4.0.2 (Scion; Bethesda MD, USA) and normalized against its respective load control (β-actin). Western blotting of each protein shown represents the mean ± S.D. of three independent experiments [14]. Blots included in Figure 2 and Figure 4 are from the same membrane. Uncropped, original membranes are shown in Appendix A. The extra band between MDAMB231 and BT549 is another breast cancer cell line not included in the final study.

### 2.4. Enzyme Activities

We evaluated enzymatic activity in breast cell whole extracts. To obtain cell extracts, cells (5–10 mg of protein) were re-suspended in 1 mL of 25 mM Tris/HCl buffer pH 7.6 plus 1 mM EDTA, 5 mM DTT, and 1 mM PMSF and disrupted by freezing in liquid nitrogen and thawing at 37 °C thrice [15]. Enzyme activities in the cellular lysate were determined at 37 °C (G6PDH, PGDH, IDH, GR, GPx, hexokinase (HK) and CS) [16] or 25 °C (CAT) [17] in 1 mL of KME buffer (120 mM KCl, 50 mM MOPS, 0.5 mM EGTA, pH 7.2). G6PDH assay contained 0.5 mM NADP^+^ and 0.05–0.15 mg of cellular protein; the reaction was started by adding 0.5 mM glucose-6-phosphate. PGDH assay was carried out in presence of 0.5 mM 6-phosphogluconate and 0.05–0.15 mg of cellular protein. IDH assay was carried out in presence of 1.2 mM MgCl_2_, 0.5 mM NADP^+^ and 0.025–0.15 mg of cellular protein; the reaction was started by adding 0.5 mM DL-isocitrate. GR assay contained 0.15 mM NADPH and 0.025–0.15 mg of cellular protein; the reaction was started by adding 1 mM GSSG. GPx assay was carried out in presence of 0.15 mM NADPH, 0.02% Triton X-100, 1U GR and 30 mM GSH; the spurious reaction was registered with the addition of 1 mM tert-butyl hydroperoxide by 20–30 s and the reaction was started by adding 0.05–0.1 mg of cellular protein. HK assay contained 0.5 mM NADPH, 1 U G6PDH, 15 mM MgCl_2_, 15 mM ATP and 0.05–0.1 mg of cellular protein; the reaction was started by adding 2 mM glucose. These assays were followed at 340 nm. CS assay was carried out in presence of 0.1 mM 3,3′-Dithiobis [6-nitrobenzoic acid] (DTNB), 0.1 mM oxaloacetate and 0.05–0.1 mg of cellular protein; the reaction was started by adding 0.1 mM acetyl-CoA following absorbance at 412 nm. CAT assay contained 5 mM H_2_O_2_; the reaction was started by adding 0.05–0.1 mg of cellular protein and followed at 240 nm. SOD assay was carried out in 1 mL of 50 mM potassium phosphate pH 7.8 plus 1 mM EDTA, 1U CAT, 0.06 mM nitro blue tetrazolium (NBT), 0.06 mg/mL albumin and 0.25 mM xanthine at 25 °C. The reaction was started by adding 15 µg of xanthine oxidase and followed at 560 nm [17]. The maximal NBT reduction obtained without protein was 100%. The amount of protein that inhibited 50% NBT reduction was defined as one unit of SOD activity (U_SOD_/mg). The range of protein assayed was 0.05–0.1 mg.

### 2.5. Determination of GSH and GSSG Levels

To obtain cellular extracts for GSH and GSG determination, the cells (2.5–5 mg of protein) suspended in PBS were mixed with ice-cold 3% (*v*/*v*) perchloric acid (PCA), mixed vigorously for at least 1 min, and centrifuged for 5 min at 20,817 g at 4 °C. The supernatant was recovered and filtered through a Millex-Millipore filter (0.45 μm pore diameter). An aliquot (0.05 mL) was analyzed by high performance liquid chromatography in a Waters 1525 system (Milford, MA, USA), using a reverse-phase C-18 Symmetry column of 3.5 μm particle size and 4.6 mm × 75 mm (Waters, Milford, MA, USA). The column was equilibrated with a buffer composed of 99% of 0.1% (*v*/*v*) trifluoroacetic acid (TFA; Sigma, St. Louis, MO, USA) in water and 1% of 100% acetonitrile (ACN, Sigma-Aldrich St. Louis, MO, USA) using a flow rate of 1 mL/min). After 10 min, a discontinuous gradient was used to obtain 80% of the TFA solution and 20% of ACN at a flow rate of 1mL/min spread over 20 min. The thiols were derivatized post-column with DTNB and detected at 412 nm in a spectrophotometer detector. Total GSH levels were determined in the absence and presence of some sodium borohydride grains to estimate GSSG levels and reduced GSH, respectively and quantified using internal standards.

### 2.6. Metabolic Profiling

Metabolic analysis was performed in a Seahorse XFe96 instrument (Agilent Technologies, Santa Clara, CA, USA) according to the manufacturer’s instructions. The following cell densities were seeded in culture plates from a Seahorse XFe FluxPak (Agilent 103792-100) per well: 20,000 MCF10A, MCF7, MDAMB231 and 10,000 for T47D and BT549. 24 h after adherence in their complete medium, the cell culture medium was replaced by DMEM XFe Medium, pH 7.4 supplemented with 10 mM glucose, 1 mM pyruvate, 2 mM glutamine, and were incubated for 45 min at 37 °C in a CO_2_ free incubator. The medium was replaced with pre-warmed DMEM XFe Medium, pH 7.4 (supplemented), and cells were incubated at 37 °C in a non-CO_2_ atmosphere until the metabolic assessment. Reagents were loaded into the sensor cartridge as indicated by the manufacturer. The Real-Time ATP Rate Assay Kit (Agilent 103592-100) was used, with the following reagent concentrations: oligomycin (oligo, 2.5 μM), rotenone/antimycin A (AA, 0.5 μM each). For the Mito stress test kit (Agilent 103015-100), oligo (2.5 μM), Carbonyl cyanide 4-(trifluoromethoxy)phenylhydrazone (FCCP, 0.5 μM) and rotenone/AA (0.5 μM each) were used for the analysis of mitochondrial function. According to the manufacturer’s instructions, ATP production and mitochondrial parameters were calculated using the Agilent Seahorse Wave Controller 2.6 software.

### 2.7. Cell Viability

Cells were treated with DL-Buthionine-(S,R)-sulfoximine (BSO, B2640, Sigma) or iodoacetate (IA, I2512 Sigma) for 48 h, or with AA (A8674, Sigma) or oligo (ALX-380-037, Enzo) for 24 h at the indicated concentrations. After treatment, cells were fixed with acetic acid (10%), methanol (10%), water (80%) for 20 min. Cells were washed with PBS and stained with crystal violet solution (0.4% crystal violet, 20% ethanol) for 20 min. Cells were washed with water and air-dried overnight. Finally, stained colonies were solubilized in acetic acid (30%) and staining was quantified at 540 nm on a Synergy (Biotek, Winooski, VT, USA) spectrophotometer.

### 2.8. Ethical Approval

Our study does not involve animal or human samples or data and thus does not require approval by an ethics committee.

### 2.9. Statistical Analysis

Graphs show mean and standard deviation (SD) of at least three independent experiments. Data were analyzed using one-way ANOVA and post hoc Turkey or Dunnett test on GraphPad Prism v5 software.

## 3. Results

### 3.1. Basal ROS Production in Breast Cancer Cell Lines

We evaluated basal ROS production via DHE staining and flow cytometry in a panel of breast cancer cell lines. We found higher ROS levels in the TN cell lines MDAMB231 and BT549. On the other hand, ER+/luminal cell lines MCF7 and T47D had lower levels of basal ROS, similar to the MCF10A non-tumorigenic cell line, which had the lowest basal levels of ROS (Figure 1).

### 3.2. Antioxidant Enzyme Protein Levels in Breast Cancer Cell Lines

The high levels of ROS in some cell lines suggested possible differences in ROS management. Thus, the levels of proteins involved in NADPH production (reducing power) and antioxidant systems were determined (Figure 2). We found differences in NADPH producing enzymes mostly in the TN cell lines but without a clear trend reflecting an increase or decrease in the pathway. The levels of G6PDH increased (3.8–5.5 fold) in BT549 cells when compared to the MCF10A or MCF7 cells. In contrast, the levels of PGDH decreased by around 95–98% in MDAMB231 and BT549 and IDH2 levels increased 2.1-fold in T47D cells when compared to MCF10A cells.

Regarding the glutathione system, no differences were observed in GR between normal and cancer cell lines. GPx1, was only detected in MCF10A and not in cancer cells. GPx4 levels increased in BT549 and MDAMB231 (1.6–2.2 fold) in comparison to MCF10A and MCF7. Regarding SOD and CAT, SOD1 protein levels increased in BT549 cells (2.1–2.6-fold) with respect to MCF10A or MCF7 cells. In MDAMB231 and BT549 cells, SOD2 increased its levels around 3.2–5-fold. Finally, CAT levels increased 2.1–2.4-fold in BT549 cells compared to MCF10A (Figure 2). Thus, the main changes observed in the protein levels of NADPH production and antioxidant systems were observed in TN cell lines with increased levels of SOD and CAT enzyme levels, elevated probably in response to their basal high levels of ROS. These results suggest that TN breast cancer cell lines might adjust their metabolism in order to keep the production of NADPH and probably suggesting a mechanism by which these cells cope with their high levels of oxidative stress.

### 3.3. Enzyme Activity of Components of NADPH-Producing Enzymes and Antioxidant Systems

In addition to protein content determination, the activity of enzymes was determined (Figure 3). In contrast to increased protein levels, the NADPH producer G6PDH, showed decreased activity (76–97%) in MDAMB231 and BT549 cells and a tendency to increase its activity (1.9-fold) in MCF7 cells compared to MCF10A cells. PGDH increased its activity (10-fold) in T47D cells and no differences were observed in the other cells when compared to MCF10A. Meanwhile, the IDH activity decreased (89–93%) in MDAMB231 and BT549 cells compared to MCF10A and MCF7. The activity of GR decreased in MDAMB231 cells compared to MCF7 cells. CAT showed a decrease in activity in T47D and MDAMB231 in comparison to MCF7. Meanwhile, SOD and GPx total activities were similar in cancer cells and non-cancer cells (Figure 3).

These results suggest a lack of correlation between protein levels and activities of components of the NADPH and antioxidant systems, as well as a decreased production of NADPH and GSH recycling in MDAMB231 and BT549 TN cells.

### 3.4. GSH and GSSG Levels in Breast Cells

The reduced (GSH) and oxidized (GSSG) glutathione levels were determined (Table 1). High levels of GSH were observed in MCF7 cells; a tendency to decrease GSH levels in T47D and MDAMB231 cells was observed when compared to MCF10A cells but this was only significant when comparing to MCF7 cells. All cancer cells showed low levels of GSH in comparison to MCF7. There was no difference between cancer and non-cancer cells in the GSSG levels. However, the GSH/GSSG ratio, which reflects the cell’s capacity to deal with oxidative stress, decreased in TN cells MDAMB231 and BT549 in comparison to MCF10A cells, probably as an effect of high levels of ROS production (Table 1), as previously suggested by metabolic profiling [18,19]. ER+ cells maintained higher levels of GSH/GSSG, reflecting their ability to cope with oxidative stress.

### 3.5. Energetic Metabolism in Breast Cancer Cell Lines

In order to evaluate possible differences between energetic metabolism and breast cancer subtypes, we evaluated the protein content of some enzymes involved in energy metabolism, such as hexokinase II (HKII) and citrate synthase (CS), as well as mitochondrial and glycolytic ATP production in the breast cancer cell line panel. The protein levels of HKII (glycolytic enzyme) and CS (TCA cycle enzyme), as well as HK activity, were similar in tumor and non-tumor cells (Figure 4). The only differences in activity were observed in CS activity which was decreased in T47D and BT549 cell lines when compared to both MCF10A and MCF7 cells (Figure 4), suggesting differences in mitochondrial metabolism in these particular cell lines. However, glycolytic ATP was increased in T47D and TN breast cancer cell lines when compared to the MCF10A non-tumorigenic control (Figure 5). Also, glycolytic ATP was increased in T47D, MDAMB231 and BT549 when compared to the MCF7 cell line (Figure 5).

Regarding mitochondrial ATP production, it was increased in ER+ breast cancer cell lines when compared to the MCF10A cells but not in the TN cell lines. When compared to MCF7 cells, MDAMB231 and BT549 cells had decreased mitochondrial ATP production (Figure 5).

### 3.6. Mitochondrial Parameters in Breast Cancer Cell Lines

Since we found differences in mitochondrial ATP production in different breast cancer cell lines, and since mitochondrial metabolism is closely related to ROS production, we evaluated several mitochondrial parameters by measuring oxygen consumption rate (OCR) with the addition of mitochondrial inhibitors to gain insight into the mitochondrial function of breast cancer cells. Basal respiration represents the oxygen consumption necessary to meet the cellular demands under baseline conditions and is calculated as the basal OCR minus non-mitochondrial respiration (OCR after oligo/FCCP and Rotenone/AA addition). In agreement with an enhanced mitochondrial ATP production in ER+ cells (Figure 5), we found an increase in basal mitochondrial respiration in MCF7 and T47D cells (Figure 6a) when compared to MCF10A cells. Also, in agreement with ATP production (Figure 5), a similar effect was observed when evaluating mitochondrial ATP production (oligomycin sensitive, Figure 6b), indicating that the ER+ cells have an enhanced mitochondrial function. In turn, the remaining basal respiration not coupled to ATP production, calculated as the basal respiration minus ATP-linked respiration and known as the proton leak, increased in all breast cancer cell lines when compared to MCF10A cells (Figure 6c), suggesting alterations in mitochondrial membrane integrity or the presence of uncoupling proteins in breast cancer cells. A similar, but inverse trend was observed when evaluating % spare respiratory capacity (Figure 6d), which indicates, as a percentage, the proportion of the basal respiration that represents the maximal respiratory capacity. It is calculated as maximal respiration (OCR after FCCP addition) divided by basal respiration × 100. These data indicate that non-tumorigenic cells have the highest percentage of spare respiratory capacity, and this parameter is reduced in cancer cells, although it did not reach statistical significance in the BT549 cell line. This parameter suggests that cancer cells either have an increased basal mitochondrial respiration (as shown on ER+ cells, Figure 6a), a decreased maximal respiration (not observed in our results), or a contribution of both parameters.

Interestingly, TN cells showed an increase in non-mitochondrial oxygen consumption (Figure 6e) when compared to MCF10A cells, calculated as the OCR that persists after adding oligomycin, FCCP, rotenone and AA. This parameter indicates non-mitochondrial enzyme activity which consumes oxygen, likely oxidases; while only MDAMB231 and BT549 had a decreased coupling efficiency (ATP production rate/basal respiration × 100) representative of what percentage of the basal respiration is used to produce ATP (Figure 6f). Thus, TN cell lines showed increased basal ROS production (Figure 1), and there might be a significant contribution of extra-mitochondrial ROS sources (Figure 6e), but also mitochondrial ROS production due to their decreased coupling efficiency (Figure 6f).

Finally, maximal respiration (OCR after FCCP addition, Figure 6g), which mimics an intense energetic demand by stimulating the respiratory chain to its maximal capacity and calculated as OCR after FCCP minus non-mitochondrial respiration, was not affected in the cancer cell lines. A similar result was observed for spare respiratory capacity in Figure 6h. Spare respiratory capacity was calculated as maximal respiration minus basal respiration and indicates the ability of the cell to respond to an energetic demand.

### 3.7. Sensitivity to GSH or Metabolic Inhibitors

Since we found important differences in GSH/GSSG ratio between the breast cancer subtypes tested and changes in metabolic regulation, we tested their sensitivity to ROS production or metabolic inhibitors. We used BSO, a GSH synthesis inhibitor, the glycolytic inhibitor IA and the mitochondrial inhibitors AA or oligo. Cells were treated with BSO or IA for 48 h or with AA or Oligo for 24 h. Results are shown in Figure 7.

In agreement with a decreased GSH/GSSG ratio found in TN cell lines, BSO, the glutathione synthesis inhibitor, selectively decreased viability of the MDAMB231 cell line, the one with the lowest GSH/GSSG ratio. However, no difference in toxicity was observed in the BT549 cell (Figure 7a), probably suggesting a delicate balance between the oxidation-antioxidant systems and a threshold for GSH/GSSG ratio since the BT549 had higher levels of this ratio when compared to the MDAMB231 cells. In agreement with the fact that glycolytic ATP was increased in most breast cancer cell lines (Figure 5), glycolysis inhibition with IA treatment decreased viability of all the BC cell lines at 5 μM (Figure 7b), indicating a vital role for this pathway in cancer progression. Finally, mitochondrial inhibition with AA or Oligo was not effective at decreasing cellular viability in any of the cell lines tested when compared to MCF10A cells. Importantly, TN cells were, in general, more resistant to mitochondrial inhibition than the other cell lines tested.

## 4. Discussion

Breast cancer is a heterogeneous disease, with the luminal A subtype and the basal or TN disease representing the subtypes with the best and worse prognoses, respectively. Breast cancer is categorized into four subtypes based on the presence or absence of hormonal receptors (ER and progesterone) or the epidermal growth factor type 2 receptor (HER2). The ER+/luminal A subtype is the most common subtype of breast cancer and its current treatment strategies include endocrine therapy and targeted therapy along with chemotherapy and radiotherapy, or with CDK4/6 inhibitors at an advanced stage [20]. TN breast cancer is characterized by the absence of these receptors (ER, progesterone receptor or HER2), and it is characterized by a higher relapse rate and a decreased survival rate 3–5 years after diagnosis. TN breast cancer is a heterogeneous subtype with six different subtypes being identified by gene expression analysis, making it harder to find a targeted therapy for this type of disease [21]. Therapies for this breast cancer subtype include mainly chemotherapy but also novel targeted drugs such as PARP inhibitors and immunotherapies [22]. With the aim of exploring metabolic vulnerabilities of these breast cancer subtypes, we analyzed ROS production, the efficiency of ROS-handling systems and energetic metabolism in a panel of breast cancer cell lines belonging to the ER+ or TN subtypes of the disease.

Previous results from our group [12] and others [23] have shown differences in basal ROS levels between cell lines representing the two subtypes of breast cancer having the best and worst prognosis for patients: the ER+ and the TN subtypes. In agreement with previously published data, we found high ROS levels, as evaluated by DHE staining and flow cytometry in the TN cell lines. On the other hand, ER+ cells had lower levels of basal ROS, similar to the non-tumorigenic control (Figure 1). These results suggest important differences in metabolism and/or ROS management systems for these breast cancer subtypes. We thus evaluated cellular energy metabolism, NADPH-producing and antioxidant systems. In general, the protein levels of components of the energy metabolism and system of ROS management were not correlated with the enzyme activity and pathway fluxes (Figure 2 and Figure 3). This may be attributed to posttranslational modifications, such as phosphorylation, ubiquitination, glycosylation, acetylation, and methylation, which may modulate protein activity [20]. For example, the activities of G6PDH and PGDH are known to be inhibited when acetylated [24].

Our analysis suggests that the high levels of ROS observed in TN MDAMB231 and BT549 cells result from the low activities of G6PDH, IDH and GR (Figure 3). Their antioxidant capacity is limited due to the low production of NADPH mediated by G6PDH and IDH, which is indispensable for the reduction of GSSG to GSH, which was also limited due to low GR activity. Therefore, the GSH/GSSG ratio was decreased in TN cells (Table 1). In agreement with this result, a previous study showed that TN cells are sensitive to inhibition of glutathione biosynthesis [19], probably because in these cells, it is challenging to maintain reduced GSH due to limitations in NADPH production and GSH reduction systems. In agreement with these observations, MDAMB231 cells, which had the lowest GSH/GSSG ratio were the most sensitive to glutathione synthesis inhibition with BSO treatment (Figure 7). However, BT549 cells, which also had a low GSH/GSSG ratio, were not sensitive to BSO treatment, indicating an important role for this balance particularly in the MDAMB231 cell and possible compensatory mechanisms in BT549 cells. Importantly, a previous study from our group [12], has shown sensitivity to glutathione precursor N-acetyl-cysteine in both cell lines, suggesting a delicate balance for glutathione ratio maintenance that when disrupted, can be toxic to TN cells.

Although malic enzyme (ME) is a producer of NADPH, in a previous study, our group reported low ME activity in comparison to IDH, G6PDH and PGDH, suggesting its low contribution to the total NADPH production in several cancer cell lines, among them the MCF7 and MDAMB231 breast cancer cells [16]. For this reason, ME was not considered in this study. In addition, IDH1 protein levels were not determined in our study. IDH1 protein content is not deferentially expressed between breast cancer cells compared to IDH2 [25]. Additionally, IDH1 contribution in the NADPH production was indirectly determined when total IDH activity was measured in the whole cell extracts because both IDH1 and IDH 2 isoforms are present. Mutations in IDH1/IDH2 might alter their contribution to NADPH production because mutated enzymes catalyze the conversion of α-ketoglutarate into D-2-hydroxyglutarate with the consumption of NADPH [26]. However, IDH mutations are common in acute myeloid leukemia, cholangiocarcinoma, chondrosarcoma, and glioma, but are rare in breast cancer [27].

The degree of oxidative stress in the cell depends on the balance between ROS generation and elimination. The high levels of ROS in TN cells may result from low antioxidant capacity and an increase in ROS production due to malfunction of the mitochondrial respiratory chain as suggested by the high proton leak observed in the absence of basal mitochondrial respiration or increased mitochondrial ATP production, or to other non-mitochondrial sources, as suggested by increased non-mitochondrial oxygen consumption in TN cells (Figure 6). These data agree with previously reported data on TN cell lines where a decrease in mitochondrial respiratory chain complex activity was described [28]. The proton leak depicts the protons that migrate into the mitochondrial matrix without producing ATP, suggestive of decreased mitochondrial membrane potential and mitochondrial ROS production [29]. In cancer cells, the expression of uncoupling proteins (UCPs), mitochondrial carriers that mediate inducible proton leak to limit ROS production, has been reported previously [30]. The high expression of UCP2 is reported to be correlated with a worse prognosis in breast cancer patients. In vitro, the inhibition of UCP2 expression in MCF-7 cells increased ROS production, apoptosis and autophagy. Therefore, UCP inhibition has been proposed as a therapeutic strategy [31], underscoring the need to evaluate UCP levels and activity in breast cancer cells, particularly in those with enhanced proton leak and no increase in total ROS, like the ER+ cells.

Interestingly, most cancer cells increased their glycolytic ATP production and ER+ cells increased their mitochondrial ATP production as well, when compared to the MCF10A non-tumorigenic cell line (Figure 5). Also, all breast cancer cell lines showed increased proton leak and most had a decreased % spare respiratory capacity, indicating certain mitochondrial damage (Figure 6). However, ER+ cells increased their basal mitochondrial respiration, mitochondrial ATP production and maintained their coupling efficiency, indicating that, although they are producing certain levels of mitochondrial ROS, their antioxidant systems compensate for this ROS production and the cells can maintain, and even increase their mitochondrial function. These results suggest an important role of NADPH-producing and glutathione-reducing systems in the maintenance of mitochondrial function in these cell lines. Alternatively, as mentioned previously, ER+ cells could have higher levels of UCP protein expression, allowing proton leak without increased mitochondrial ROS production.

Breast cancer cell lines were more sensitive to IA treatment than the MCF10A non-tumorigenic control. IA, a glycolytic inhibitor, acting primarily on gluceraldehyde-3-phosphate dehydrogenase, but with other functions described as the inhibition of G6PDH [32] induced a decrease in cellular viability particularly at 5 μM (Figure 7). These data agree with increased glycolytic activity and sensitivity to glycolysis inhibitors in cancer cells [3,33], and highlights the opportunity to tackle this pathway for cancer therapy. In contrast, TN cells were more resistant to mitochondrial inhibitors than MCF10A or ER+ cells, consistent with a possible mitochondrial damage characterized by increased proton leak and decreased mitochondrial coupling efficiency (Figure 6). Importantly, increased migration in cancer has been shown to occur in combination with mitochondrial dysfunction, thus underscoring the need to test mitochondrial inhibitors in breast cancer cell migration and invasion.

The data in this work indicate that TN breast cancer cells maintain a delicate balance between ROS-producing and ROS-scavenging systems. Although we do not identify a causal relationship, metabolic adaptations in energetic metabolism and PPP occurring in TN breast cancer cells might be related to maintaining high levels of mitochondrial and extra-mitochondrial ROS, which might be useful for sustaining their proliferative and invasive potential. It will be interesting to test other non-mitochondrial ROS sources in TN breast cancer cells to find specific targets for therapeutic intervention. Finally, our results show that a comprehensive understanding of metabolic changes occurring in different cancer subtypes is a feasible method to uncover metabolic vulnerabilities which can be targeted for cancer therapeutics. In this sense, this study might propose combinatory therapy for TN breast cancer affecting glycolysis and GSH metabolism.

## 5. Conclusions

Breast cancer is a heterogeneous disease, with subtypes like the TN disease having the worst prognoses among all breast cancer subtypes despite recent advances in therapies. Our study indicates important metabolic and ROS-management differences among breast cancer subtypes and highlights important vulnerabilities in these pathways. We found a general dependence on glycolysis despite different glycolytic ATP levels in breast cancer cell lines; and high levels of ROS in TN cells. These high ROS levels are maintained in TN cells because of limited antioxidant capacity in the NADPH-producing and GSH systems, mitochondrial dysfunction and non-mitochondrial ROS production, increasing their sensitivity to GSH synthesis inhibitors. Our data indicate that metabolic and antioxidant profiling of breast cancer will provide important targets for metabolic inhibitors or antioxidant treatment in breast cancer therapy.

## Figures and Tables

**Figure 1 metabolites-14-00435-f001:**
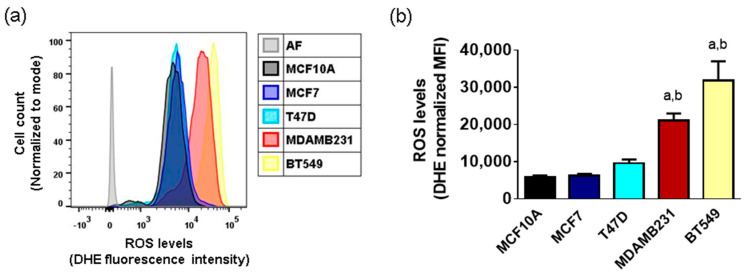
ROS levels in a panel of breast cancer cell lines. Basal ROS production was measured with DHE staining and flow cytometry in non-tumorigenic (MCF10A) cells, ER+/luminal (MCF7 and T47D) and TN (MDAMB231 and BT549) breast cancer cell lines. In (**a**), data show one representative experiment of 3–4 different preparations. AF, autofluorescence. (**b**) shows mean + SD of 3–4 independent experiments. MFI, mean fluorescence intensity; a, *p* < 0.05 vs. MCF10A non-tumorigenic cell line; b, *p* < 0.05 vs. MCF7, ER+ cell line. One-way ANOVA, Tukey post hoc.

**Figure 2 metabolites-14-00435-f002:**
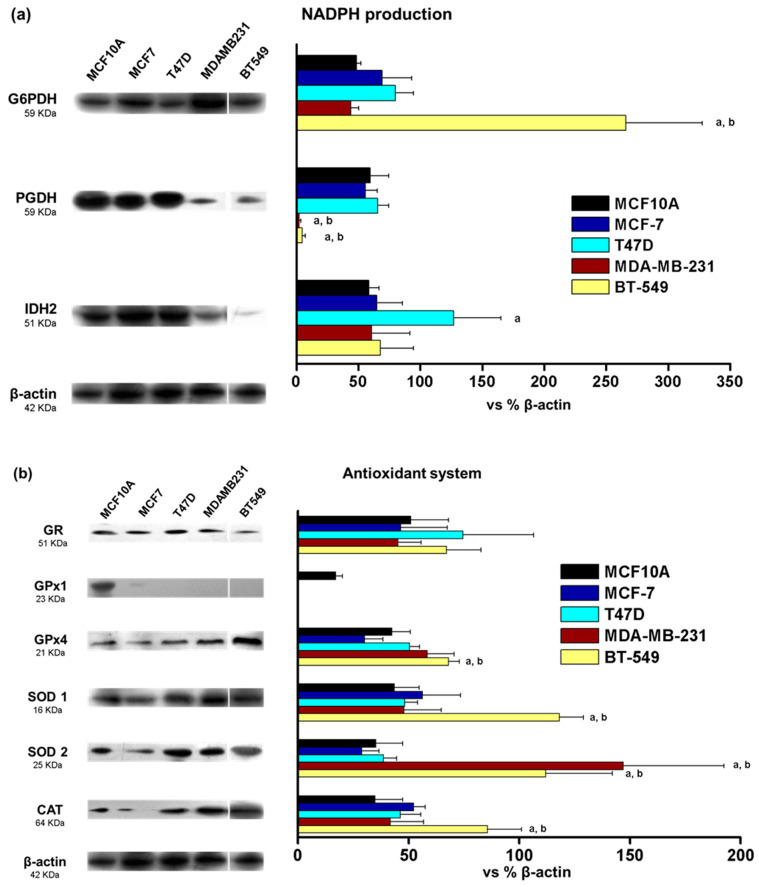
Protein levels of NADPH-producing enzymes (**a**) and proteins from the antioxidant system (**b**) from normal (MCF10A) and tumor (MCF7, T47D, MDAMB231 and BT549) breast cancer cell lines. Bar graphs show normalization against β-actin. The data shown represent the mean ± S.D. of three different preparations. G6PDH, glucose-6-phosphate dehydrogenase; PGDH, 6-phosphogluconate dehydrogenase; IDH2, isocitrate dehydrogenase 2; GR, glutathione reductase; GPx, glutathione peroxidase; SOD, superoxide dismutase; CAT, catalase. ANOVA/post hoc Tukey analysis. a represents *p* < 0.05 vs. MCF10A; and b represents *p* < 0.05 vs. MCF7. Blots included in the figure are from the same membrane; for more details, see Section 2.3.

**Figure 3 metabolites-14-00435-f003:**
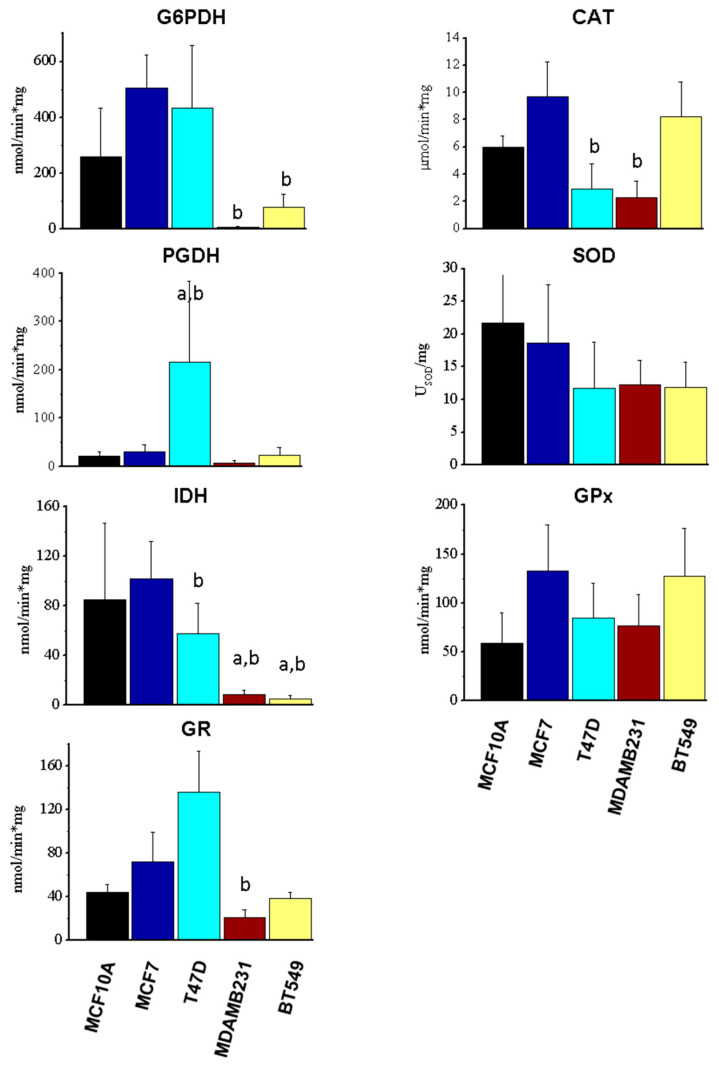
Activities of enzymes participating in NADPH production and antioxidant systems. The data shown represent the mean ± S.D. of 3–5 different preparations. G6PDH, glucose-6-phosphate dehydrogenase; PGDH, 6-phosphogluconate dehydrogenase; IDH, isocitrate dehydrogenase; GR, glutathione reductase; GPx, glutathione peroxidase; SOD, superoxide dismutase; CAT, catalase ANOVA/Tukey post hoc analysis with a representing *p* < 0.05 vs. MCF10A; and b representing *p* < 0.05 vs. MCF7.

**Figure 4 metabolites-14-00435-f004:**
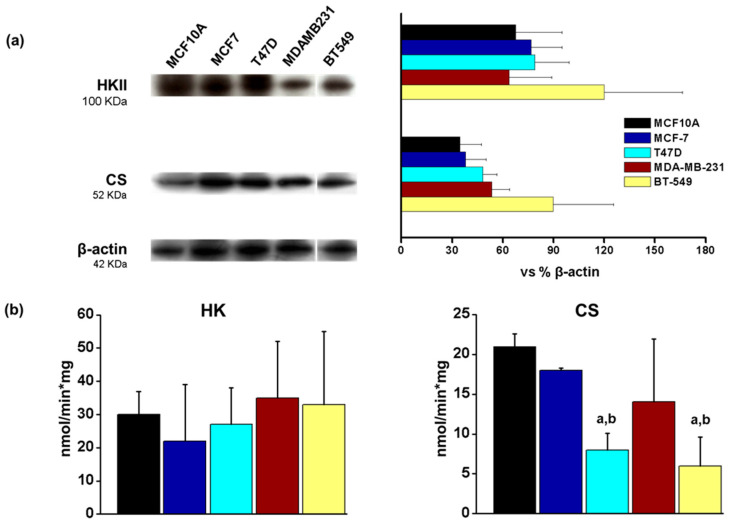
Protein levels and activity of hexokinase (HK) and citrate synthase (CS). (**a**) Histograms represent data normalized to β-actin levels, the data show mean ± S.D. of three different preparations. (**b**) Activities of HK and CS. The data show the mean ± S.D. of 3–5 different preparations. ANOVA/post hoc Tukey analysis with a representing *p* < 0.05 vs. MCF10A and b, *p* < 0.05 vs. MCF7. Blots included in the figure are from the same membrane; for more details, see Section 2.3.

**Figure 5 metabolites-14-00435-f005:**
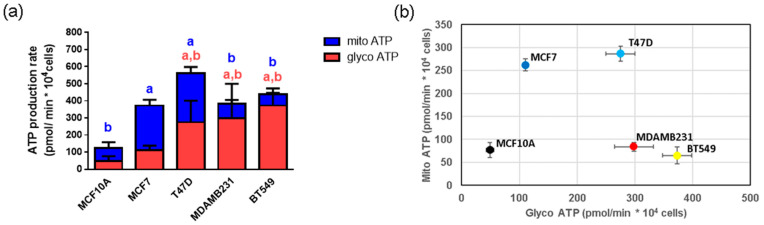
Mitochondrial and Glycolytic ATP production in a panel of breast cancer cell lines. ATP production was measured in non-tumorigenic (MCF10A) cells, ER+ (MCF7 and T47D) and TN (MDAMB231 and BT549) breast cancer cell lines. In (**a**), data show mean +/− SD from 3–7 independent experiments. a shows differences with respect to MCF10A and b with respect to MCF7 (*p* < 0.05), blue and red letters represent differences in mitochondrial (mito) ATP and glycolytic (glyco) ATP, respectively. In (**b**), the same data are shown, using mean ± SEM for clarity. One-way ANOVA, Dunnett’s post hoc.

**Figure 6 metabolites-14-00435-f006:**
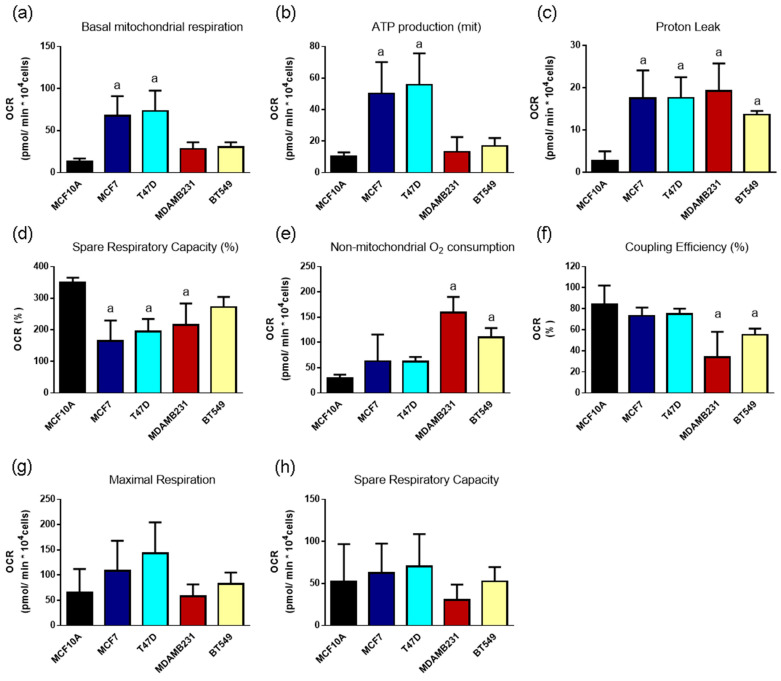
Mitochondrial parameters evaluated in a panel of breast cancer cell lines, including a non-tumorigenic control (MCF10A), ER+ (MCF7 and T47D) and TN (MDAMB231 and BT549) breast cancer cell lines. (**a**) Basal mitochondrial respiration, (**b**) ATP production, (**c**) Proton leak, (**d**) Spare respiratory capacity, (**e**) Non-mitochondrial O_2_ consumption, (**f**) Coupling efficiency, (**g**) Maximal respiration and (**h**) Spare respiratory capacity. Data show mean ± SD of 3–4 independent experiments performed in replicates. Each graph mentions the mitochondrial parameter evaluated; a represents differences with *p* < 0.05 when compared to MCF10A non-tumorigenic control. One-way ANOVA, Dunnett’s post hoc.

**Figure 7 metabolites-14-00435-f007:**
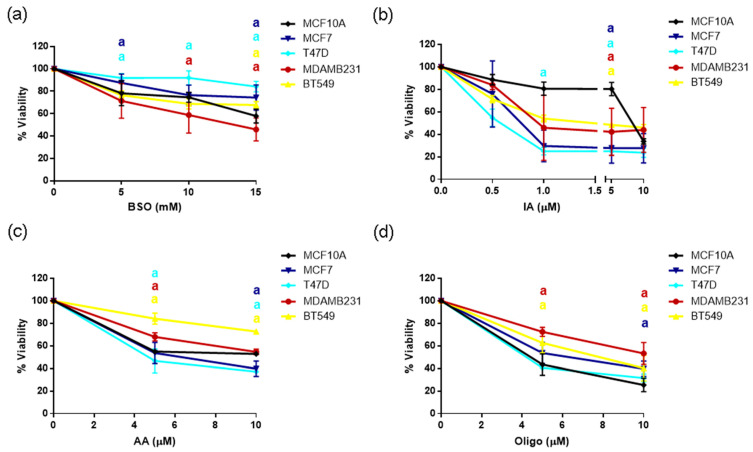
Sensitivity to ROS-inducing agents, or metabolic inhibitors: (**a**) BSO, buthionine sulfoximine; (**b**) IA, iodoacetate; (**c**) AA, antimycin A; (**d**) oligo, oligomycin A in the non-tumorigenic control (MCF10A), ER+ (MCF7 and T47D) and TN (MDAMB231 and BT549) breast cancer cell lines. Data show mean ± SD of 3–4 independent experiments performed in replicates. a *p* < 0.05 when compared to MCF10A non-tumorigenic control. Two-way ANOVA, Dunnett’s post hoc.

**Table 1 metabolites-14-00435-t001:** Levels of GSH and GSSG in breast cell lines.

Cell Line	GSH [nmol/mg prot]	GSSG [nmol/mg prot]	GSH/GSSG
MCF10A	18.4 ± 7.8	0.19 ± 0.17	97
MCF7	54 ± 14 ^a^	0.85 ± 0.47	63
T47D	4.9 ± 3.2 ^b^	0.15 ± 0.1	33
MDAMB231	5.8 ± 2.3 ^b^	0.5 ± 0.46	12
BT549	13.8 ± 4.3 ^b^	0.8 ± 0.54	17

ANOVA/post hoc Tukey analysis with ^a^ *p* < 0.05 vs. MCF10A; ^b^ *p* < 0.05 vs. MCF7.

## Data Availability

The original contributions presented in the study are included in the article/Appendix A, further inquiries can be directed to the corresponding authors.

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
