# Peer review of "Metabolic and Oxidative Stress Management Heterogeneity in a Panel of Breast Cancer Cell Lines"

_metabolites, 2024, doi:10.3390/metabo14080435_

Round 1

Reviewer 1 Report

Comments and Suggestions for Authors

The manuscript by Maycotte is of scientific sound, but numerous improvements are required before considering the manuscript suitable for publication.

·        Many abbreviations are not explicit. Please spell out the terms the first time they appear in the text or create a list of abbreviations.

·        In section 2.2, please provide the catalog numbers of antibodies used.

·        I suggest rearranging the methods in the order they appear in the manuscript. For example, ROS measurement before Western Blotting, and so on.

·        There are errors in the numbering in the methods section (Cell viability and statistical analysis).

·        Figure 1 is representative of 3 preparations. For greater clarity, I ask you to create a graph with the average values ​​and standard deviation resulting from the 3 independent experiments.

·        In Figure 2 some western blot membrane cropping is not acceptable, e.g. G6PDH, PGDH, IDH2, SOD1 and β-actin. The lanes relative to BT549 were cut from other membranes and added, please indicate this.

·        The graph in Figure 2 is confusing both for the colors used and for how the standard deviations are indicated. Please improve.

·        Other enzymes generate NADPH: IDH1 and Malic enzyme 1 (ME1). They must be investigated to have a complete understanding of the mechanism. The expression of IDH1 must also be evaluated to understand whether it is differentially expressed within the different cell types and in comparison to IDH2.

·        Furthermore, IDH1/ 2 mutations are frequent in different tumors, and this should be discussed.

·        Line 247: correct TD47D

·        Fig S1 could be part of the main manuscript.

·        The statistical analysis reported in Figures 2-3-4-6 is incomprehensible. Different colors, unclear symbols: the legend often doesn't help. I invite you to revise the representation of statistical analysis in figures to make them understandable. To indicate statistical differences I suggest using letters, where different letters indicate significance.

Comments on the Quality of English Language

Minor editing of English language required

Reviewer 2 Report

Comments and Suggestions for Authors

In my opinion, the article by Maycotte et al. is solidly written and meets the high scientific standards. The research problem was well presented and all assumptions and hypotheses were implemented. Nevertheless, I would like to ask the authors to make a few minor corrections and respond to a few comments:

1. The work contains minor editorial errors, for example chapters 2.1., 2.5. and several others are not justified

2. The description of the methods used contains errors in the numbering of subsequent subchapters; after chapter 2.6. Metabolic profiling appears in the chapter titled 2.4. Cell viability and 2.5 Statistical analysis.

3. Chapter 3.2. Antioxidant enzyme protein levels in breast cancer cell is very difficult to read. The authors briefly describe the increase/decrease of individual proteins in a single sentence, but in my opinion it is done in a rather chaotic way.

4. The chart in Figure 2 is not very legible, it is difficult to understand what the form of notation a and b means

5. Why do some of the wells in Western Blot photos (for example, for the CAT protein and the BT549 line) look like they were glued on and came from another gel?

6. In the case of GSH or GSSG level determinations and the data contained in Table 1, I suggest placing the unit of concentration of the mentioned proteins directly in the table, for example GSH [nmol/mg]. However, this is only a suggestion.

7. Request to improve the quality of the supplement, especially the captions of the chart axes and the chart itself.

Reviewer 3 Report

Comments and Suggestions for Authors

I would like to thank the authors for their  impressive work showing the difference in oxidative stress responses of estrogen receptor positive  and triple negative breast cancer.

The main purpose, method, results and discussion of the study are very well written.

I have a few suggestions regarding the article.

1- I think you do not need to share your results in the last paragraph of the introduction.

2- I recommend that the information shared in the first paragraph of the results section, including references 16-17, be evaluated in the discussion section instead of the results.

3- The expression ''triple negative'' is abbreviated as ''TN'' in the introduction and repeated as triple negative (TN) in the results and discussion sections. It will be sufficient to show the abbreviation only in the first usage.

Kind regards

Reviewer 4 Report

Comments and Suggestions for Authors

The paper titled- Metabolic and oxidative stress management heterogeneity in a panel of breast cancer cell lines represents an interesting study about enzymes involved in cellular metabolism in breast cancer cell lines reported to non-malignant cell lines that represents a critical point to testing the Warburg theory. Still, your manuscript needs significant revisions to increase its quality, and I suggest the following changes:

1-     Please revise the entire abstract to highlight your study's objectives, importance, and novelty in correlation with the obtained results, discussion, and conclusion. Also, specify a few words about your methods integrated into the context.

2-     Please revise all sentences between 90 to 100 lines in the Introduction section. Specify the importance and vision of your study because your observations about the cell metabolism in cancer cell lines must be described in the Results and Discussion sections. Please replace “Here” with “Our study” to highlight your objectives.

3-     In the Materials and Methods section, please revise and describe the malignant and non-malignant breast cell lines and used media with more attention, specifying the manufacturer. Please specify how you obtain the cancer cell extract.

4-     To the ROS measurement from Materials and Methods, please add the fluorochrome and channel because you studied more intracellular superoxide enzyme than total cellular oxidative stress. In cell viability analysis, please define if you are applying a treatment to cancer cells (you wrote- after treatment?). You may also use the flow cytometry method to study cell viability. How do you describe the fold by metabolic profile? Please revise all the methods and statistical presentation to be more accessible.

5-     In the Results section, please present and interpret your obtained results by media and standard deviation (X±SD). In Figure 1, please modify the explanation of axes; you must write the ROS count using fluorochrome on Y-axe and samples (cell lines) on X-axe.

6-     Please revise all results from this section to be more accessible and explain your obtained results.

7-     Please add conclusions and ethical approval for your study.

Comments on the Quality of English Language

Please revise the English style and integrate sentences on the page in conformity with the journal request.

Round 2

Reviewer 1 Report

Comments and Suggestions for Authors

The authors responded satisfactorily and appropriately edited the manuscript. No more issue was detected. The manuscript can be accepted in this form

Comments on the Quality of English Language

English language is quite fine. 

Reviewer 2 Report

Comments and Suggestions for Authors

Thank you for taking all of my comments and suggestions into consideration.

Reviewer 4 Report

Comments and Suggestions for Authors

The manuscript was revised in conformity with reviewer requisition.